# An expanded cell wall damage signaling network is comprised of the transcription factors Rlm1 and Sko1 in *Candida albicans*

**Marienela Y. Heredia[1], Mélanie A. C. Ikeh[2], Deepika Gunasekaran[2,3], Karen A. Conrad[1], Sviatlana Filimonava[1], Dawn H. Marotta[1], Clarissa J. Nobile[2], Jason M. Rauceo[1]***

**1** Department of Sciences, John Jay College of the City University of New York, New York, New York, United States of America, **2** Department of Molecular and Cell Biology, School of Natural Sciences, University of California Merced, Merced, California, United States of America, **3** Quantitative and Systems Biology Graduate Program, University of California Merced, Merced, California, United States of America

* jrauceo@jjay.cuny.edu

**Data Availability Statement:** All relevant data are within the manuscript and its Supporting Information files.

## Abstract

The human fungal pathogen *Candida albicans* is constantly exposed to environmental challenges impacting the cell wall. Signaling pathways coordinate stress adaptation and are essential for commensalism and virulence. The transcription factors Sko1, Cas5, and Rlm1 control the response to cell wall stress caused by the antifungal drug caspofungin. Here, we expand the Sko1 and Rlm1 transcriptional circuit and demonstrate that Rlm1 activates Sko1 cell wall stress signaling. Caspofungin-induced transcription of *SKO1* and several Sko1-dependent cell wall integrity genes are attenuated in an *rlm1Δ/Δ* mutant strain when compared to the treated wild-type strain but not in a *cas5Δ/Δ* mutant strain. Genome-wide chromatin immunoprecipitation (ChIP-seq) results revealed numerous Sko1 and Rlm1 directly bound target genes in the presence of caspofungin that were undetected in previous gene expression studies. Notable targets include genes involved in cell wall integrity, osmolarity, and cellular aggregation, as well as several uncharacterized genes. Interestingly, we found that Rlm1 does not bind to the upstream intergenic region of *SKO1* in the presence of caspofungin, indicating that Rlm1 indirectly controls caspofungin-induced *SKO1* transcription. In addition, we discovered that caspofungin-induced *SKO1* transcription occurs through self-activation. Based on our ChIP-seq data, we also discovered an Rlm1 consensus motif unique to *C. albicans*. For Sko1, we found a consensus motif similar to the known Sko1 motif for *Saccharomyces cerevisiae*. Growth assays showed that *SKO1* overexpression suppressed caspofungin hypersensitivity in an *rlm1Δ/Δ* mutant strain. In addition, overexpression of the glycerol phosphatase, *RHR2*, suppressed caspofungin hypersensitivity specifically in a *sko1Δ/Δ* mutant strain. Our findings link the Sko1 and Rlm1 signaling pathways, identify new biological roles for Sko1 and Rlm1, and highlight the complex dynamics underlying cell wall signaling.

**Funding:** This research was funded by the National Institutes of Health (NIH) National Institute of General Medical Sciences (NIGMS)(https://www.nigms.nih.gov), grant number SC3GM111133 to J. M.R. and NIGMS award R35GM124594 and by a Pew Biomedical Scholar Award from the Pew Charitable Trusts to C.J.N. This work was also supported by the Kamangar family in the form of an endowed chair to C.J.N. The funders had no role in study design, data collection and analysis, decision to publish, or preparation of the manuscript.

**Competing interests:** The authors have declared that no competing interests exist.

## Author summary

*Candida albicans* is the most common human fungal pathogen isolated in clinical settings. The echinocandin drug caspofungin is used to treat invasive candidiasis; however, the emergence of increasing echinocandin resistance underscores the need for new antifungal strategies. Elucidating the signaling mechanisms that govern caspofungin-induced tolerance has the potential to identify candidate proteins that could serve as novel therapeutic targets. Here, we expand the Rlm1 and Sko1 cell wall transcriptional network and find that Rlm1 indirectly regulates Sko1 signaling. Furthermore, we identify Sko1- and Rlm1-specific biological roles in caspofungin adaptation, such as osmoregulation and secretion. Lastly, we discover a protective role for glycerol in caspofungin tolerance. Overall, these findings provide mechanistic insight into the genetic and cellular bases underlying cell wall signaling in *C. albicans*.

## Introduction

Fungal signaling pathways are critical for environmental interactions and survival. The yeast *Candida albicans* colonizes diverse biotic and abiotic surfaces in the human host and causes superficial and life-threatening infections in immunocompromised patients [1, 2]. *C. albicans* encounters rapidly changing microenvironments caused by fluctuating pH, temperature, osmolarity, $O_2$ depletion, resident microbiota, host innate defenses, and antifungal drugs [3]. These conditions all modulate the cell wall. As a fungal-specific structure, the cell wall is required for adhesion, morphogenesis, and resistance to plasma membrane turgor pressure [4]. Transcription factors are a critical component of signaling pathways, as they control genes involved in developmental processes, stress adaptation, and pathogenesis. An understanding of cell wall signaling mechanisms at the transcriptional level can identify prospective genes, which can be valuable in developing antifungal interventions.

The echinocandin class of antifungal drugs causes cell lysis by inhibiting β-1-3 glucan synthesis [5]. *C. albicans* executes a robust transcriptional program in response to cell wall damage (CWD) caused by the echinocandin caspofungin [6–8]. To date, the transcription factors Cas5, Rlm1, and Sko1 have been shown to mediate caspofungin-induced cell wall stress signaling. Deletion of *CAS5* results in hypersensitivity to various cell wall and plasma membrane perturbants and abrogates pathogenicity *in vivo* [7, 9, 10]. Cas5 is dephosphorylated by Glc7 in response to CWD and regulates gene classes whose functions include cell wall biogenesis and integrity and cell cycle control [8]. Although Cas5 is a major regulator of CWD response in *C. albicans*, Cas5 does not contain a homolog in the model yeast *Saccharomyces cerevisiae*.

Rlm1 is the key transcriptional regulator of CWD signaling in *S. cerevisiae* and is a downstream target of the Protein Kinase C (PKC) mitogen-activated protein kinase (MAPK) signaling pathway [11]. In *C. albicans*, an *rlm1Δ/Δ* mutant strain is hypersensitive to caspofungin and shows reduced virulence in murine infection models [7, 12]. Despite this, Rlm1 was shown to regulate a small fraction of caspofungin-responsive genes, and its activators remain unknown [7]. Recently, Rlm1 was shown to play a role in cell wall remodeling and filamentation in *C. albicans* when yeast-form cells were grown in lactate as a carbon source [13].

Sko1 function has been extensively characterized in *S. cerevisiae*. In response to osmotic stress, Sko1 mediates adaptation via the MAPK signaling cascade known as the High Osmolarity Glycerol (HOG) pathway [14]. Sko1 is phosphorylated by the MAP kinase Hog1 to both activate and repress its target genes [15]. Prior gene expression and biochemical analyses revealed that Sko1 is conserved as a regulator of osmotic stress signaling in *C. albicans*, but

there is considerable divergence in its target genes [6, 16]. Sko1 regulation of the CWD response is independent of HOG pathway signaling and requires the glucose-partitioning kinase Psk1 [6]. Both *sko1Δ/Δ* and *psk1Δ/Δ* mutant strains are hypersensitive to caspofungin. Sko1 also regulates hypoxic and oxidative stress signaling and hyphal formation [17, 18]. The diverse processes controlled by Sko1, Rlm1, and Cas5 underscore their versatility and importance in *C. albicans* biology.

Here we have utilized molecular and cellular approaches to characterize the Sko1 and Rlm1 CWD signaling circuit. We identify new gene classes and biological processes that are regulated by Sko1 and Rlm1. Finally, we demonstrate the interconnected relationship between Sko1 and Rlm1 to coordinate the CWD response.

## Results

### Regulation of *SKO1* transcription by Rlm1

*SKO1* transcription is significantly upregulated in a Psk1-dependent manner following caspofungin exposure [6], but a direct activator of Sko1 has not yet been identified. Microarray analysis has previously reported that Rlm1 partially controls *SKO1* transcription in nutrient-rich medium [12]; however, it is unknown whether Rlm1 controls *SKO1* transcription in response to CWD. We hypothesized that caspofungin-induced *SKO1* expression may require Rlm1 and/or Cas5. We considered Cas5 because it is the only other known regulator of caspofungin-induced signaling, and *SKO1* was not included in the microarray dataset of a *cas5Δ/Δ* mutant strain treated with caspofungin [7]. We performed RT-qPCR to measure S*KO1* transcript levels in wild-type, *cas5Δ/Δ* mutant, and *rlm1Δ/Δ* mutant strains. As was previously observed [6], *SKO1* expression significantly increased following caspofungin treatment in wild-type cells (Fig 1A). *SKO1* expression was relatively unchanged in the *cas5Δ/Δ* mutant strain under basal conditions and was slightly greater following caspofungin treatment compared to the wild-

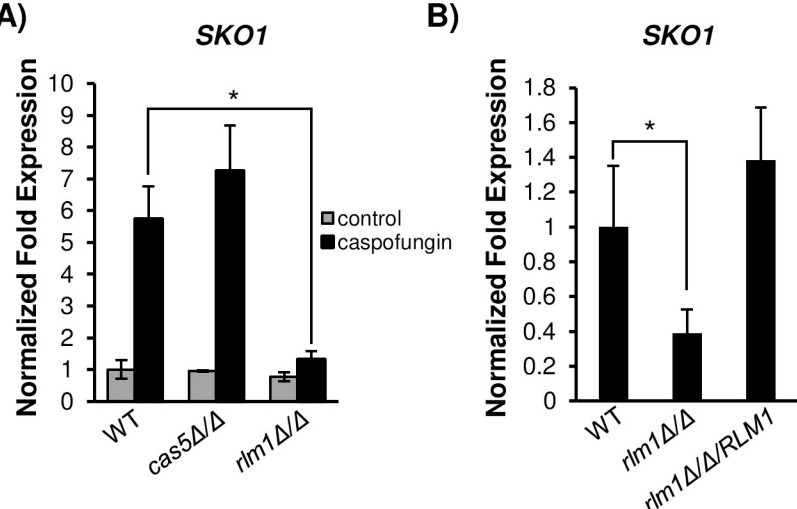

**Fig 1. RT-qPCR analysis of *SKO1* expression in transcription factor mutants.** A) *SKO1* expression was measured in the wild-type strain (WT), *cas5Δ/Δ* mutant strain, and *rlm1Δ/Δ* mutant strain. All strains were treated with 125 ng/mL caspofungin or dH₂O (control) for 30 min. Transcript levels were normalized to *TDH3* (which encodes glyceraldehyde-3-phosphate dehydrogenase) expression, and fold changes were normalized to the untreated wild-type strain adjusted to the value of 1.0. B) *SKO1* expression was measured in the wild-type strain, *rlm1Δ/Δ* mutant strain, and *rlm1Δ/Δ/RLM1* complemented strain, and fold changes were normalized to the caspofungin-treated wild-type strain adjusted to the value of 1.0. Data presented represents the mean of three biological replicates. The asterisk indicates a *P*-value of ≤ 0.05 between the wild-type and *rlm1Δ/Δ* mutant strains.

type strain; however, this increase was not significant. This finding establishes that *SKO1* is not a downstream target of Cas5.

Strikingly, *SKO1* expression was significantly reduced in the *rlm1Δ/Δ* mutant strain treated with caspofungin (Fig 1A) and, in concordance with previous microarray experiments, slightly reduced in the untreated *rlm1Δ/Δ* mutant strain (Fig 1A, [12]). To confirm that this phenotype was due to the loss of Rlm1 function, we measured *SKO1* expression in an *rlm1Δ/Δ/RLM1* complemented strain following caspofungin treatment. Our findings show that reintroduction of a single *RLM1* allele in the *rlm1Δ/Δ* mutant strain is sufficient to restore *SKO1* expression to wild-type levels (Fig 1B). Taken together, our gene expression findings show that Rlm1 partially controls caspofungin-induced *SKO1* expression.

The connection between *SKO1* and *RLM1* implicates an expanded role for Rlm1 in caspofungin-induced cell wall signaling than previously reported [7]. If Rlm1 controls CWD signaling in part by activating *SKO1*, we then reasoned that known Sko1-dependent CWD genes must also require Rlm1 for full expression. We measured the expression of *PHR2*, *KRE9*, *KRE1*, *MNN2*, and *SKN1* in *rlm1Δ/Δ* mutant cells following caspofungin treatment. These genes were chosen based on their known cell wall functions [19]. Also, previous microarray and RT-qPCR results showed significantly reduced transcription of these genes in the *sko1Δ/Δ* mutant strain following caspofungin treatment [6]. Our RT-qPCR results show that expression of these genes was reduced at least 1.5 fold in the *rlm1Δ/Δ* mutant strain compared to the wild-type strain (Fig 2). Therefore, Rlm1 also partially controls Sko1-dependent genes in CWD signaling.

## Identification of Rlm1 direct gene targets

Although previous microarray analysis showed that Rlm1 only regulated two caspofungin-induced genes [7], our RT-qPCR results indicate that Rlm1 regulates additional CWD response genes (Figs 1 and 2). It is unknown, however, whether these genes are direct targets of Rlm1. To address this, we performed genome-wide ChIP-seq experiments to identify direct Rlm1 target genes. We analyzed Rlm1 upstream intergenic region enrichment in the presence or absence of caspofungin using a wild-type strain containing an Rlm1-V5 tagged protein. We found Rlm1 enrichment at a total of 30 gene upstream intergenic regions. Of these, 5 are potential false positives due to their association with genomic regions that are highly enriched in ChIP-seq experiments irrespective of antibody specificity [20]; these 5 genes were excluded from further analyses. We observed upstream intergenic region enrichment for 18 genes in response to caspofungin and 4 genes under untreated conditions. 3 genes showed equal levels of upstream intergenic region enrichment under both conditions (S1 Appendix and S2 Table).

Interestingly, Rlm1 directly bound to the upstream intergenic regions of key cell wall integrity genes including *PGA59*, *PGA56*, and *PGA62*, as well as the cell wall glycosidase-encoding gene, *PHR2* (Fig 3A and 3B and S1 Appendix). *PHR2* is significantly upregulated in response to caspofungin [7]. However, in that study, *PHR2* did not require Rlm1 for full expression. Also, strains containing mutations to the *PHR2* paralogous gene, *PHR1*, were found to be hypersensitive to caspofungin [21]. Although we show that Rlm1 controls transcription of *SKO1* and Sko1-dependent CWD genes (Figs 1 and 2), we did not identify *SKO1* as a direct gene target of Rlm1. Therefore, based on our results, Rlm1 appears to be an indirect regulator of *SKO1* transcription.

Next, we compared our Rlm1 ChIP-seq dataset against existing *C. albicans* Rlm1 microarray datasets to determine if Rlm1 directly binds to and regulates transcription of known downstream target genes. We identified two uncharacterized directly bound Rlm1 target genes, *ORF19.3698* and *ORF19.5322*, from our ChIP-seq dataset that were identified as differentially

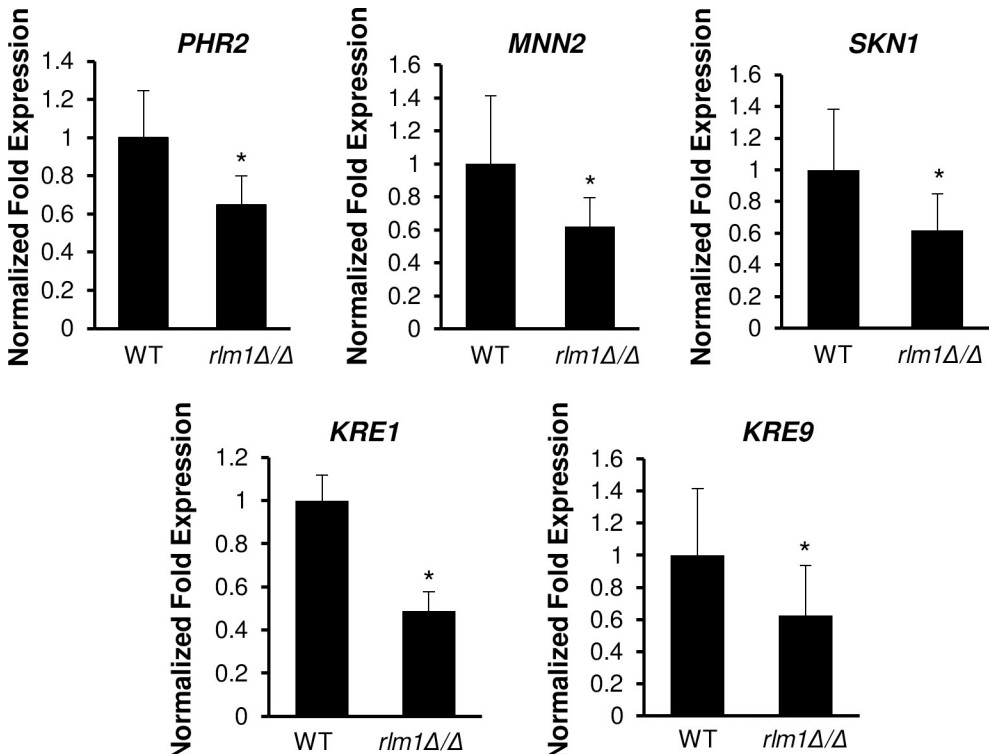

**Fig 2. RT-qPCR analysis of Sko1-regulated caspofungin-responsive genes.** Gene expression was monitored in *C. albicans* wild-type and *rlm1Δ/Δ* mutant strains following 30 minutes of 125 ng/mL caspofungin treatment. Data presented represents the mean of three biological replicates. Transcript levels were normalized to *TDH3* expression, and fold changes between strains were normalized to the wild-type reference strain adjusted to a value of 1.0. The asterisk indicates a *P*-value of ≤ 0.05 between the wild-type and *rlm1Δ/Δ* mutant strains.

regulated in the absence of any stressors in the *rlm1Δ/Δ* microarray dataset from Delgado-Silva et al. [12]. Surprisingly, we did not identify Rlm1 directly bound target genes that matched the Bruno et al. caspofungin microarray dataset [7]. We note that our experiments were performed under similar growth conditions as both microarray experiments. Taken together, our ChIP-seq results reveal a new set of directly bound Rlm1-dependent target genes that were previously undetected by DNA microarrays.

We performed Cluster of Orthologous Groups (COG) and Gene Ontology (GO) analyses with our ChIP-seq dataset to annotate target genes and to identify new Rlm1 biological roles. We also utilized the Kyoto Encyclopedia of Genes and Genomes (KEGG) to identify pathways for Rlm1 target genes. COG analysis showed that uncharacterized genes accounted for the greatest percentage (32%) of Rlm1 direct gene targets (S1 Fig and S2 Table). Other important categories included intracellular trafficking (8%), and translation (8%). GO enrichment for the cellular component category showed that several Rlm1 direct target genes encode for proteins that localize to cytoskeletal sites such as microtubules and the spindle pole (Fig 3F). GO enrichment for the biological processes category implicated Rlm1 in protein and organelle localization and secretion (Fig 3E). Notable direct gene targets include the secretory vesicle component-encoding gene *SEC10* (S1 Appendix), and the glucose sensor-encoding gene, *SHA3* (Fig 3C). KEGG analysis did not identify any pathways associated with Rlm1 direct target genes. Taken together, our results identify a new class of Rlm1 dependent caspofungin-responsive directly bound target genes including genes with functions in cell wall integrity. Furthermore, we identify a potential role for Rlm1 in regulating molecular localization and secretion.

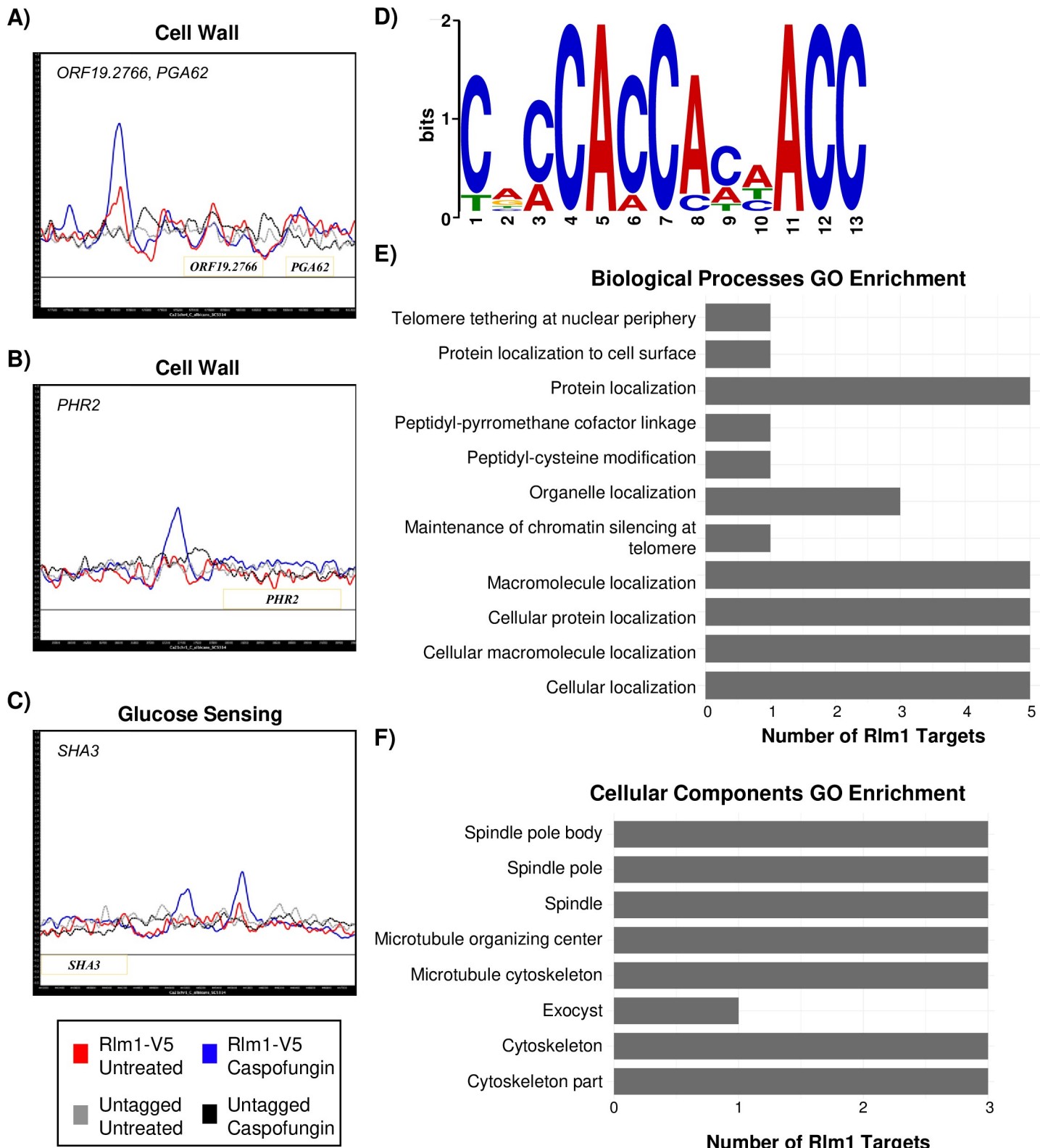

**Fig 3. Identification of Rlm1 direct target genes via ChIP-seq.** Wild-type cultures containing epitope-tagged Rlm1-V5 were treated with or without 125 ng/mL caspofungin for 30 minutes prior to immunoprecipitation. Immunoprecipitation was also conducted using untagged wild-type caspofungin-treated cells to examine non-specific binding with the anti-V5 antibody. The ChIP-seq binding data was mapped and plotted using MochiView and the Rlm1 target genes A) *ORF19.2766* and

*PGA62*, B) *PHR2*, and C) *SHA3* are shown. Blue peaks correspond to caspofungin-treated Rlm1-V5 cells, and red peaks correspond to untreated Rlm1-V5 cells. The black dashed line represents untagged Rlm1 cells treated with caspofungin, and the grey dashed line represents untagged Rlm1 untreated cells. The X-axis represents 4000 bp ORF chromosomal locations, and the Y-axis represents enriched coverage compared to the expected average coverage. Genes (yellow boxes) plotted above the bold line are read in the sense direction, and genes below the line are read in the antisense direction. D) A 500 bp region around the top enriched binding targets was used to identify the binding motif recognized by Rlm1 using MEME and corroborated using MochiView. E-F) GO enrichment analyses for Biological Processes and Cellular Components (*P*-value < 0.01) for all Rlm1 target genes are shown.

### Identification of Sko1 direct gene targets

Previous microarray and RT-qPCR gene expression analyses showed that Sko1 regulates the transcription of 26 caspofungin-induced genes [6]. We performed ChIP-seq analysis to identify Sko1 direct target genes. Similar to our Rlm1 experiments, we utilized a wild-type strain containing a Sko1-V5 tagged protein [6] to determine Sko1 upstream intergenic region enrichment in the presence or absence of caspofungin. We found Sko1 enrichment at 108 gene upstream intergenic regions, of which 21 were potential false positives (based on [20]) and thus excluded from further analyses (S2 Appendix and S2 Table). We observed enhanced upstream intergenic region enrichment for 85 genes in response to caspofungin and for one gene under basal conditions (S2 Appendix and S2 Table). Interestingly, we found that Sko1 was enriched at its own upstream intergenic region and shows higher levels of enrichment during caspofungin treatment (Fig 4G and S2 Appendix). In addition, Sko1 appears to bind across its own ORF irrespective of caspofungin treatment (Fig 4G and S2 Appendix). Although the basis of this binding is unclear, one possibility is that Sko1 is involved in activating transcription elongation. Alternatively, Sko1 ORF binding could be due to an artifact of the ChIP-seq procedure, which has been observed to occur for genes that are highly actively transcribed [22]. Based on these findings, *SKO1* transcription may be, at least partly, dependent on self-activation.

We compared our Sko1 ChIP-seq dataset against existing microarray datasets examining transcriptional changes in the *sko1Δ/Δ* mutant strain exposed to caspofungin [6] or sodium chloride [16] to assess whether Sko1 direct binding modulates transcription. We included the sodium chloride dataset because osmotic stress causes cell wall remodeling [23]. We identified a total of 30 genes from our ChIP-seq data that were previously shown to be regulated by Sko1 (S2 Table). Specifically, in response to caspofungin, we identified 6 known Sko1 caspofungin-induced genes (out of the 26 caspofungin-induced gene targets identified in previous microarrays), including *PGA31* (Fig 4D), and 13 known Sko1 caspofungin-repressed genes (S2 Table). We identified 17 Sko1-activated genes and 4 Sko1-repressed genes in response to osmotic stress, including the alcohol dehydrogenase-encoding gene, *ADH1* (S2 Table), and the glycerol metabolic enzyme-encoding genes, *RHR2* (Fig 4H) and *GPD2* (S2 Table). Collectively, these results demonstrate that Sko1 directly binds to and regulates a group of major stress response genes.

Because the majority of Sko1 directly bound target genes were not identified in previous *sko1Δ/Δ* microarrays, we performed COG, GO, and KEGG analyses to gain insight into the physiological and molecular roles of Sko1 target genes. COG analysis revealed the majority of Sko1 target genes (17%) were uncharacterized. Major categories for Sko1 target genes include signal transduction (7%), post-translational modification (7%), carbohydrate transport and metabolism (7%), intracellular trafficking and secretion (3%), and transcription (6%) (S1 Fig and S2 Table).

Our GO enrichment findings for biological processes revealed that Sko1 regulates osmotic stress response target genes upon caspofungin treatment (Fig 4K). Notable target genes include the sodium transporter-encoding gene, *ENA21* (Fig 4F), and the HOG pathway two-component phosphotransferase regulator-encoding gene, *YPD1* (S2 Appendix). GO enrichment

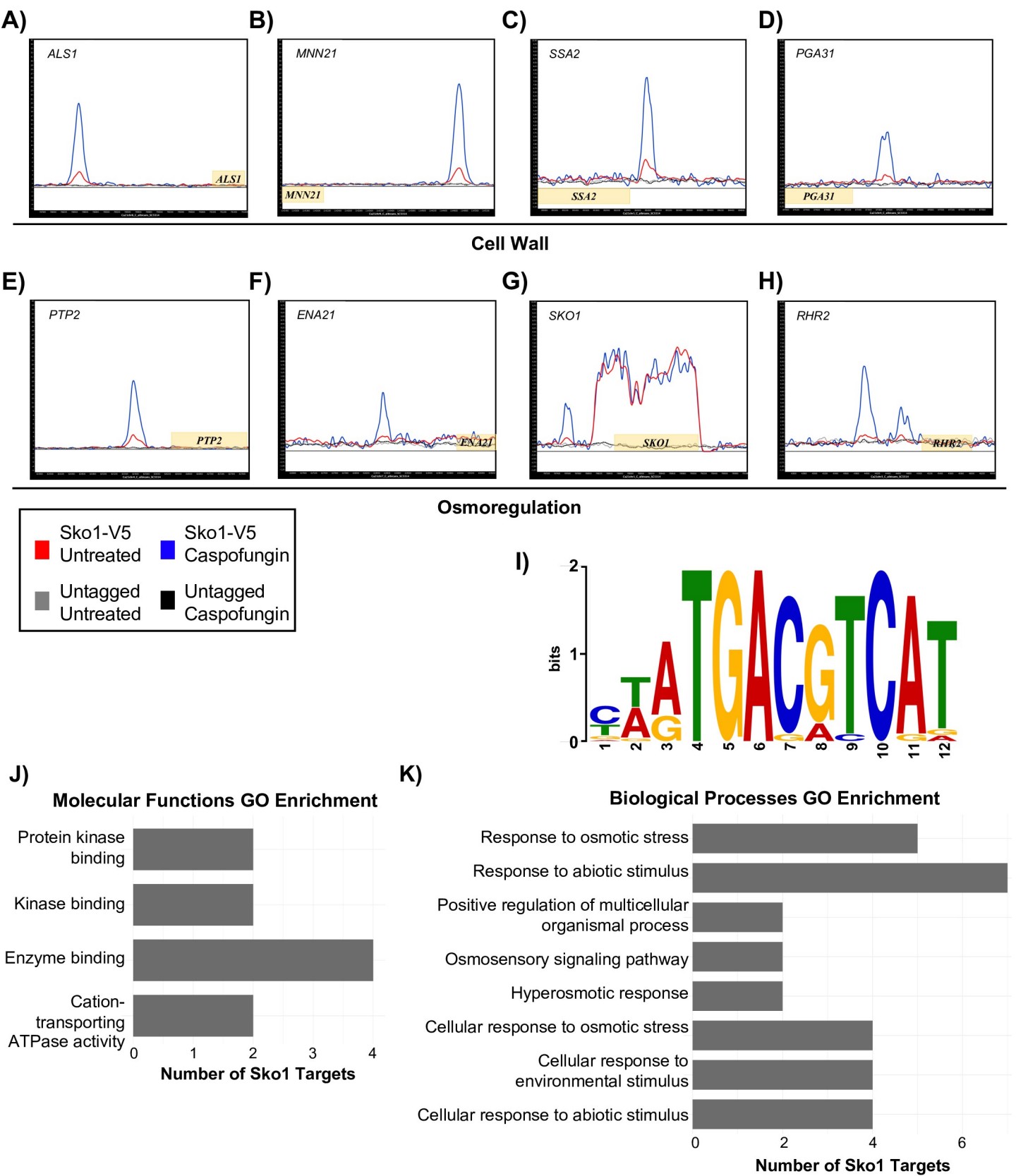

**Fig 4. Identification of Sko1 direct target genes via ChIP-seq.** Wild-type cultures containing epitope-tagged Sko1-V5 were treated with or without 125 ng/mL caspofungin for 30 minutes prior to immunoprecipitation. Immunoprecipitation was also conducted using untagged wild-type caspofungin-treated cells to examine non-specific binding with the anti-V5 antibody. The ChIP-seq binding data was mapped and plotted using MochiView and the Sko1 gene targets A) *ALS1*, B) *MNN21*, C) *SSA2*, D) *PGA31*, E) *PTP2*, F) *ENA21*, G) *SKO1*, and H) *RHR2* are shown. Blue peaks correspond to caspofungin-treated Sko1-V5 cells, and red peaks correspond to untreated Sko1-V5 cells. The black dashed line represents untagged Sko1 cells treated with caspofungin, and the grey dashed line represents untreated Sko1 untreated cells. The X-axis represents 4000 bp ORF chromosomal locations, and the Y-axis represents enriched coverage compared to the expected average coverage. Genes (yellow boxes) plotted above the bold line are read in the sense direction, and genes below the line are read in the antisense direction. I) A 500 bp region around the top enriched binding targets was used to identify the binding motif recognized by Sko1 using MEME and corroborated using MochiView. J-K) GO enrichment analyses for Molecular Functions and Biological Processes (*P*-value < 0.01) for all Sko1 target genes are shown.

findings for the molecular functions category highlighted genes associated with protein kinase binding (Fig 4J). We did not find any enrichment for the cellular localization category. KEGG analysis identified several Sko1 targets that are essential in regulating MAPK stress signaling pathways including the phosphatase-encoding gene, *PTP2* (Fig 4E and S3 Fig). Collectively, our ChIP-seq findings and computational analyses identify a new set of Sko1 target genes and reveal a new role for Sko1 in caspofungin-induced osmoregulation.

## Identification of a Sko1 and Rlm1 DNA-binding consensus motif

Previous *in silico* analyses of our osmotic stress microarray dataset identified two putative Sko1 DNA binding motifs [16]. The consensus motif, (A/T)ATAGCAAT(T/C)A, was identified for genes repressed by Sko1 and the motif, (T/C)TCATCTCATC(G/T)CA(A/T), for genes activated by Sko1. For Rlm1, *in silico* promoter analysis has not been performed on microarray datasets; however, Rlm1 target genes were previously screened for the presence of the *S. cerevisiae* Rlm1 binding motif CTAWWWWTAG [12, 24].

We utilized the motif-detecting software MEME and the Motif Finder function in Mochi-View to identify authentic Rlm1 and Sko1 DNA consensus motifs based on direct binding events from our ChIP-seq data. For Rlm1, we identified the consensus motif, CACCACCA-CAACC ($P = 1.5e^{-05}$ and IC score of 19), and found that 1–8 instances of the motif occurred in the upstream intergenic regions of directly bound Rlm1 target genes (Fig 3D, S2 Table). Overall, the Rlm1 motif we identified was found approximately 2 times more frequently in the upstream intergenic regions of directly bound Rlm1 target genes compared to the frequency of finding it in intergenic regions of the entire *C. albicans* genome. Interestingly, the *C. albicans* consensus motif we identify here is distinct from that of the Rlm1 consensus motif identified in *S. cerevisiae*, indicating that the Rlm1 consensus binding motifs between these two fungal species are highly diverged. Although the function of Rlm1 in cell wall signaling is conserved in both species, this finding exemplifies the prevalence of transcriptional rewiring in *C. albicans*. Overall, our promoter analyses reveal a novel Rlm1 DNA-binding consensus motif for *C. albicans*.

For Sko1 target genes, we identified the consensus motif, CTATGACGTCAT ($P = 2.9e^{-20}$ and IC score 18.2), from our ChIP-seq data, where 1–3 instances of the motif occurred in the upstream intergenic regions of directly bound Sko1 target genes (Fig 4I, S2 Table). Overall, the Sko1 motif was found approximately 4 times more frequently in Sko1 target genes compared to the frequency of finding it in intergenic regions of the entire *C. albicans* genome. Notably, the Sko1 consensus motif identified here by ChIP-seq bears some similarity to the motif for Sko1-activated genes previously identified by microarray studies. We utilized the Yeast Transcription Factor Specificity Compendium (YeTFaSCo) database [25] to determine whether the *C. albicans* Sko1 motif was similar to known transcription factor DNA-binding motifs in *S. cerevisiae*. We found that our sequence partially matched the *S. cerevisiae* Sko1 ATF/CREB consensus motif, T(G/T)ACGT(C/A)A. This result indicates that the Sko1 DNA binding motif is likely conserved between *S. cerevisiae* and *C. albicans*.

## *SKO1* suppression analyses

Overexpression suppression analyses can be used to determine if two functionally related genes are part of the same pathway [26]. Since Rlm1 partially controls *SKO1* transcription and the expression of several Sko1 dependent genes (Figs 1 and 2), we hypothesized that *SKO1* overexpression would suppress the caspofungin hypersensitivity phenotype of the *rlm1Δ/Δ* mutant strain. We overexpressed *SKO1* in an *rlm1Δ/Δ* mutant strain and a wild-type strain and monitored cell growth in the presence of caspofungin. To verify *SKO1* overexpression, we monitored transcription of the Sko1 and Rlm1 caspofungin-dependent genes *PGA13*, *PGA31*, and *CRH11* [6, 7] in the $P_{TDH3}SKO1$-*rlm1Δ/Δ* overexpressing strain. *CRH11*, *PGA13*, and *PGA31* expression was significantly elevated in the *SKO1* overexpressing *rlm1Δ/Δ* mutant strain compared to the wild-type and *rlm1Δ/Δ* mutant strains (S2 Fig). Thus, our *SKO1* overexpressing strain is functional and significantly increases transcription of *SKO1* and *RLM1* target genes.

Our spot plate growth assays showed that the *SKO1* overexpressing wild-type strain had no apparent phenotype under basal growth conditions or in the presence of caspofungin (Fig 5A). The *rlm1Δ/Δ* mutant strain grew poorly in the presence of 50 ng/mL caspofungin, consistent with observations from previous studies [7, 12]. Strikingly, growth was partially restored in the *rlm1Δ/Δ* mutant strain overexpressing *SKO1* (Fig 5A). In addition, our spot plate results were consistent with our microplate liquid kinetic growth assays (Fig 5B). Collectively, these results, combined with our RT-qPCR results (Figs 1 and 2), indicate that the Sko1 and Rlm1 cell wall stress signaling pathways are linked in *C. albicans*.

## Role of glycerol metabolism in *SKO1* suppression

Glycerol metabolism is elevated in wild-type cells treated with caspofungin, presumably to counteract osmotic stress caused by CWD [6, 12, 27, 28]. We hypothesized that overexpression of *SKO1* alleviates caspofungin hypersensitivity in the *rlm1Δ/Δ* mutant strain in part by upregulating the glycerol synthesis gene, *RHR2*. We overexpressed *RHR2* in both *rlm1Δ/Δ* and *sko1Δ/Δ* mutant strains and examined growth in the presence of caspofungin. *RHR2* overexpression in the *sko1Δ/Δ* mutant strain partially rescued caspofungin hypersensitivity (Fig 6A). This finding implies that in a *sko1Δ/Δ* mutant strain, CWD-induced osmotic stress contributes to caspofungin hypersensitivity. On the other hand, *RHR2* overexpression did not suppress caspofungin hypersensitivity in the *rlm1Δ/Δ* mutant strain (Fig 6B). Thus, while *SKO1* overexpression is sufficient to suppress caspofungin hypersensitivity in the *rlm1Δ/Δ* mutant strain (Fig 5), the mechanism governing this phenotype is independent of glycerol metabolism.

## Discussion

Caspofungin is the premier echinocandin antifungal drug for combating invasive infections caused by *C. albicans* [5]. Transcription factors Sko1, Rlm1, and Cas5 control the adaptive response to caspofungin-induced CWD. Although the molecular mechanisms governing Cas5 signaling have been characterized [7, 8], several key biological questions remain unanswered regarding Sko1 and Rlm1 function. Here, we show that the Sko1 and Rlm1 cell wall stress signaling pathways are interconnected and introduce a new set of Rlm1 and Sko1 directly bound downstream target genes, several of which, are uncharacterized in *C. albicans* (Fig 7). Our findings define the Sko1 and Rlm1 genetic response to caspofungin-induced cell wall stress and underscore the importance of signal rewiring and pathway redundancy to fine-tune the adaptive response to CWD in *C. albicans*.

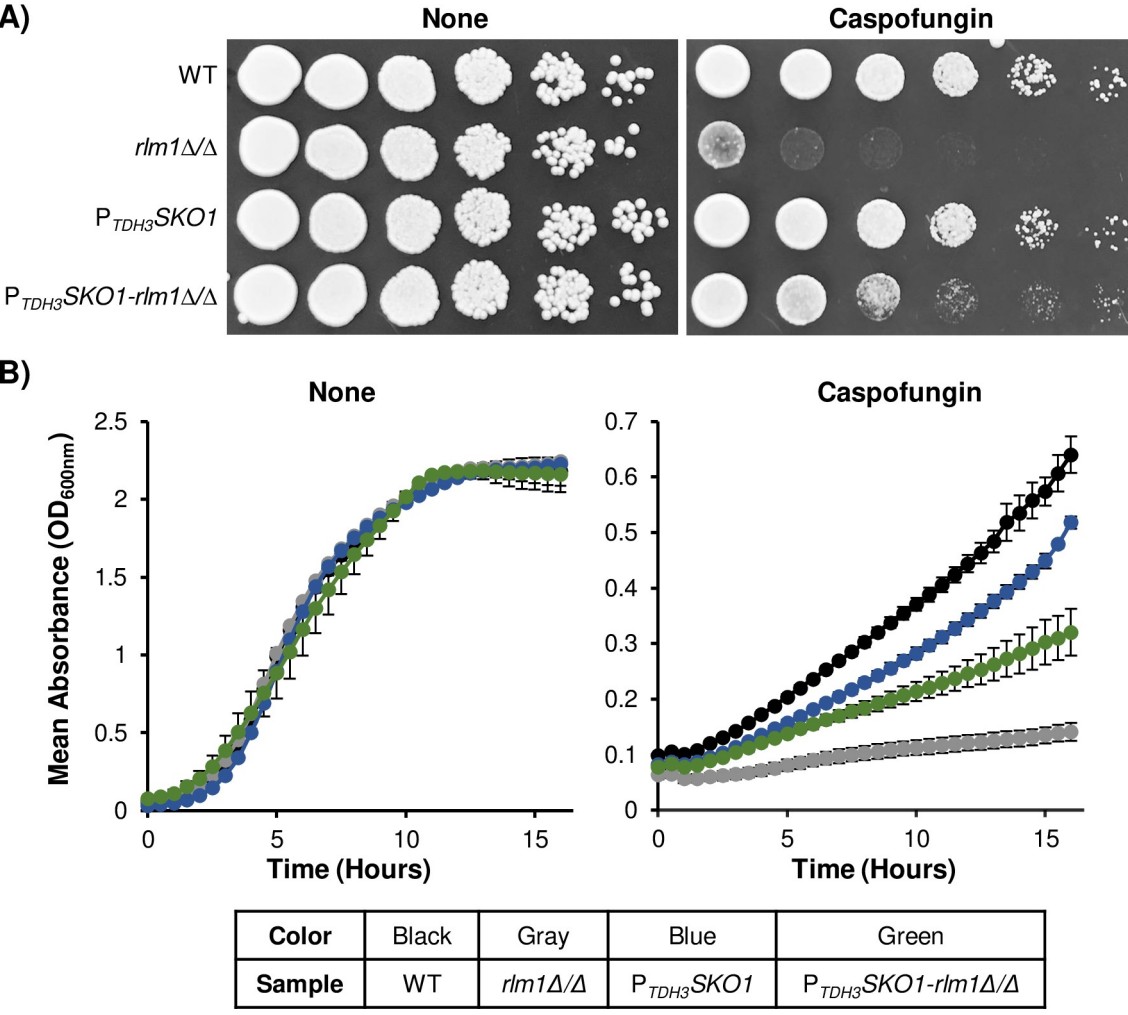

**Fig 5. Characterization of *SKO1* overexpression.** Solid and liquid growth assay results of the wild-type strain, *rlm1Δ/Δ* mutant strain, $P_{TDH3}SKO1$ overexpressing strain, and $P_{TDH3}SKO1$-*rlm1Δ/Δ* overexpressing strain. A) Overnight cultures were serially diluted onto nutrient YPD$^{uri}$ plates supplemented with or without 50 ng/mL caspofungin. Plates were photographed following 48 hours of growth at 30˚C. B) Growth kinetics were monitored in a microplate reader. Overnight cultures were standardized to an OD$_{600nm}$ of 0.2 in 200 μL YPD$^{uri}$ and incubated with water as a control or caspofungin. For each assay, three biological replicates were analyzed. Experiments were repeated two times, and data presented depicts one representative experiment. The line graph on the left represents untreated samples, and the graph on the right represents samples treated with 50 ng/mL caspofungin.

## Crosstalk between Sko1 and Rlm1 cell wall signaling

A major goal motivating this study was to determine the regulatory mechanisms underlying Sko1 activation of CWD signaling. Our gene expression and phenotypic suppression findings demonstrate that Rlm1 regulates *C. albicans* caspofungin-induced CWD signaling in part through the activation of Sko1 signaling. Four major lines of evidence support this hypothesis: (i) *SKO1* caspofungin-induced expression is reduced over 5-fold in an *rlm1Δ/Δ* mutant strain (Fig 1); (ii) several Sko1-specific caspofungin-response genes showed reduced expression in an *rlm1Δ/Δ* mutant strain (Fig 2); (iii) *SKO1* overexpression specifically alleviated caspofungin hypersensitivity in an *rlm1Δ/Δ* mutant strain (Fig 5); and (iv) prior gene expression results showed that Rlm1 activates *SKO1* transcription in the absence of caspofungin treatment [12]. In *S. cerevisiae*, the PKC and HOG MAP kinase pathways coordinate CWD signaling following exposure to the β-1-3 glucanase, zymolyase. Rlm1, but not Sko1, is required to mount the

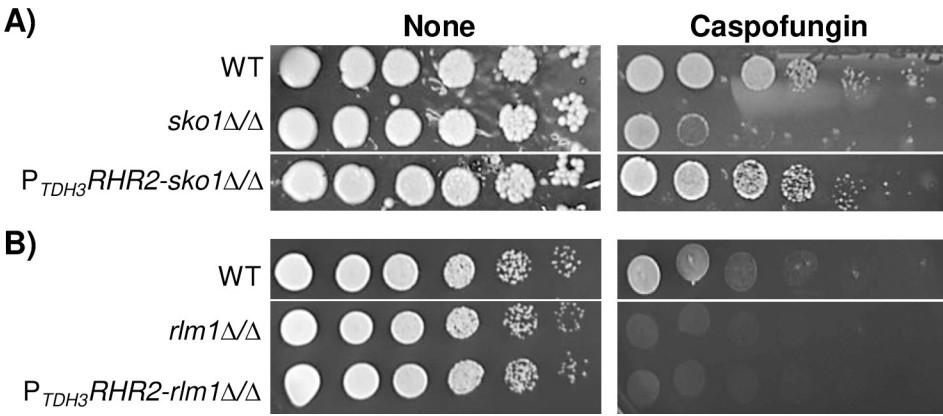

**Fig 6. Glycerol overproduction analysis.** Growth assay results of the wild-type strain, *sko1Δ/Δ* mutant strain, *rlm1Δ/Δ* strain, $P_{TDH3}RHR2$-*sko1Δ/Δ* overexpressing strain and $P_{TDH3}RHR2$-*rlm1Δ/Δ* overexpressing strain. Overnight cultures were serially diluted onto nutrient YPD$^{uri}$ plates supplemented with or without 50 ng/mL caspofungin. Plates were photographed following A) 48 and B) 24 hours of growth at 30°C. Images were cropped to exclude samples not relevant to this study.

transcriptional response to zymolyase [29]. Thus, the Rlm1-Sko1 connection represents a new paradigm in transcriptional regulation of CWD in *C. albicans*.

## Role of Rlm1 direct gene targets

The cellular and molecular bases of caspofungin hypersensitivity for cell wall transcriptional regulators has been enigmatic, where multiple diverse biological processes are known to be

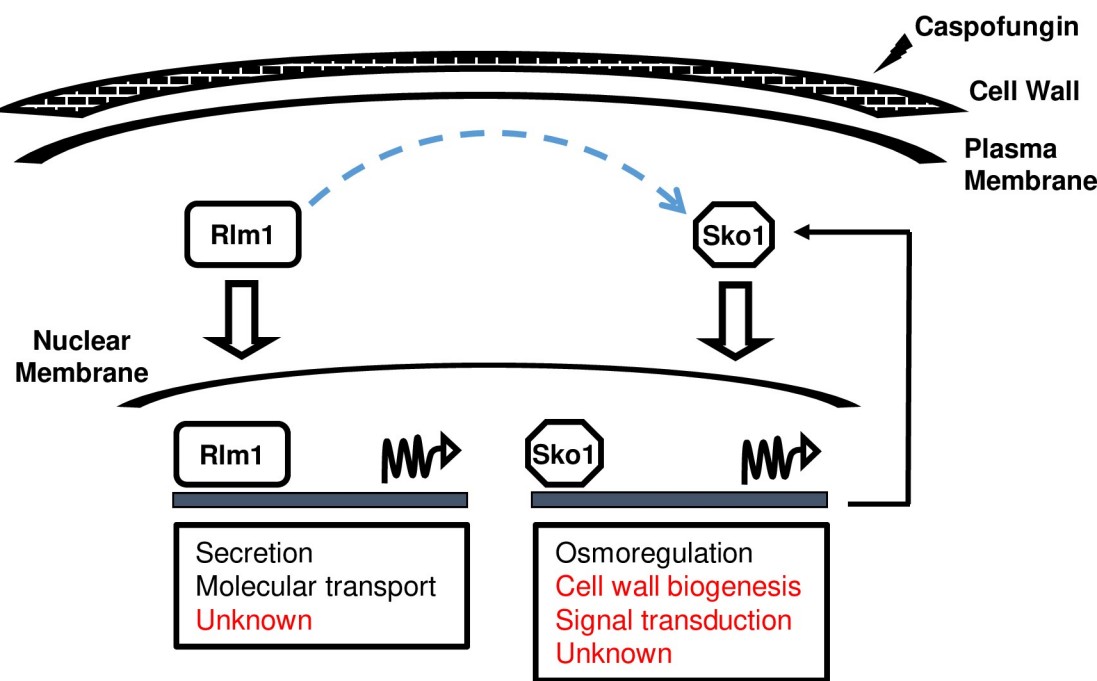

**Fig 7. Rlm1/Sko1 cell wall damage signaling.** Following caspofungin treatment (lightning symbol), Rlm1 and Sko1 regulate diverse gene classes (boxed) to restore cell wall integrity. Gene classes identified using GO enrichment are in black font, and gene classes identified using COG analysis are in red font. Increased *SKO1* expression occurs directly through Sko1 self-activation (bent solid arrow). Rlm1 indirectly activates *SKO1* transcription by an unknown post-translational mechanism (dashed blue arrow).

affected by CWD, thus leading to presumable pleiotropic effects [6–8]. Following caspofungin stress, a previous gene expression microarray study found that Rlm1 modulated the expression of only 5 genes, with two genes (*CRH11* and *PGA13*), being necessary for cell wall integrity [7]. Although the *pga13Δ/Δ* and *crh11Δ/Δ* mutants are hypersensitive to several cell wall perturbants [30, 31], *pga13Δ/Δ* is only slightly sensitive to caspofungin [6]. Further, *PGA13* and *CRH11* are upregulated by Sko1 and Cas5 [6, 7]. Taken together, it seems likely that other factors contribute to caspofungin hypersensitivity. Another previous gene expression microarray study in the absence of caspofungin treatment found that the *rlm1Δ/Δ* mutant strain had altered expression of 773 genes (including several transcriptional regulators, *SKO1*, and Sko1-dependent genes) [12]. Therefore, the simplest explanation for the caspofungin hypersensitivity of the *rlm1Δ/Δ* mutant strain is that the mutant is unable to activate Sko1 signaling. Our gene expression and growth suppression results support this hypothesis (Figs 1, 2 and 5). We further addressed this hypothesis by performing genome-wide ChIP-seq experiments.

Our ChIP-seq dataset identified 25 Rlm1 direct target genes, which did not include *SKO1* or any known transcriptional regulator (S1 Appendix and S2 Table). Only two uncharacterized genes from our ChIP-seq dataset were previously identified by Rlm1 microarray experiments [12], highlighting the relevance of ChIP-seq experiments to study gene regulation. We examined the amino acid sequence of all 25 proteins encoded by these target genes [19] for the presence of a putative DNA-binding domain but failed to identify any candidates. Our upstream intergenic region analyses of the Rlm1 target genes identified a novel *C. albicans* Rlm1 binding consensus motif, CACCACCACAACC. While we find that Rlm1 does not directly regulate Sko1 or another transcription factor, our results identify a new class of direct Rlm1 target genes that could be important players in the CWD response.

Our COG analysis shows that most Rlm1 target genes are uncharacterized (S1 Fig). However, Rlm1 localizes to the upstream intergenic regions of four key cell wall genes: *PGA56*, *PGA59*, *PGA62*, and *PHR2* (Fig 3A and 3B and S1 Appendix). Similar to the *rlm1Δ/Δ* mutant strain, *pga59Δ/Δ*, *pga62Δ/Δ*, and *phr2Δ/Δ* mutant strains are sensitive to cell wall perturbants and contain increased levels of chitin in their cell walls [32, 33]. Increased chitin synthesis is characteristic of a possible compensatory mechanism to maintain cell wall integrity [34] and occasionally confers caspofungin resistance as observed with the *pga62Δ/Δ* mutant strain [21, 32]. We argue that the attenuated regulation of *PGA59*, *PGA62*, and *PHR2* could lead to reduced integrity of the *rlm1Δ/Δ* mutant cell wall.

Our GO biological process and cellular component enrichment category results indicate that Rlm1 regulates macromolecular and organelle localization, where several Rlm1-dependent target genes are predicted to associate with microtubules. Notable genes include two essential genes: *SEC10* and *SHA3*. *SEC10* encodes a protein that is a component of the highly conserved exocyst complex, which mediates polarized secretion [35, 36]. In *S. cerevisiae*, the Rlm1 regulator Slt2 is required for polarized cell growth [37, 38]; however, it is unknown whether Slt2 mediates cell growth via Rlm1. Our findings reveal a potential new role for Rlm1 function in *S. cerevisiae*. *SHA3* encodes a kinase involved in glucose transport and sensing [36, 39]. Further, a *sha3Δ/SHA3* heterozygous mutant displays altered colony morphology. The cell wall is primarily composed of carbohydrates, and thus interruption of Sha3 and Sec10 function may have profound consequences for biological processes requiring cell wall function.

## Context-dependent Sko1 control of cell wall-induced osmotic stress

Transcriptional networks in *C. albicans* display a high level of redundancy where multiple transcription factors can regulate one another as well as overlapping sets of target genes [17, 40]. Consequently, it is often difficult to determine the contributions of each individual

transcriptional regulator in an adaptive response that is mediated by multiple regulators within a complex transcriptional network. We employed genome-wide ChIP-seq to define the individual role of Sko1 in CWD signaling. From these results, we identified 87 direct Sko1 target genes (S2 Appendix). Our upstream intergenic region analyses of the Sko1 target genes identified the consensus motif, CTATGACGTCAT for *C. albicans* Sko1. This sequence is similar to the canonical *S. cerevisiae* Sko1 ATF/CREB motif [14], highlighting conserved aspects of Sko1 function in *C. albicans*.

Similar to Rlm1, our COG analysis revealed that the majority of Sko1 target genes are uncharacterized. Our GO enrichment and KEGG analyses, however, revealed that a subset of the Sko1 target genes is required for the osmotic stress response. Major "hallmark" genes for this response include *SOU1*, *ENA21*, *YPD1*, *GPD2*, *RHR2*, and *HGT19* (Fig 4 and S2 Appendix); these genes are significantly upregulated in wild-type strains following NaCl treatment and are regulated by the Hog1 MAP kinase [41]. *RHR2*, *GPD2*, and *HGT19* require Sko1 for full expression following osmotic stress [16]. An *rhr2Δ/Δ* mutant strain is sensitive to cationic stress [28], and heterologous expression of *C. albicans ENA21* in *Candida dubliniensis* suppresses a hyperosmotic growth defect [42].

*ENA21*, *GPD2*, *RHR2*, *SOU1*, and several *HGT* family members are significantly modulated in response to caspofungin stress. Sko1 and Cas5 partially regulate *GPD2* and *RHR2* [6, 8]. However, our ChIP-seq data demonstrates that Sko1 directly regulates *YPD1*, *HGT19*, *ENA21*, and *SOU1*. Although we previously identified *GPD2* and *RHR2* as caspofungin-induced Sko1 targets from microarray experiments [6], the failure to identify additional osmotic stress-responsive genes in that prior study led us to conclude that Sko1 was not involved in coordinating the osmotic stress response generated by caspofungin treatment. Our identification of additional osmotic stress hallmark genes and HOG pathway regulators as direct Sko1 target genes in this study, however, establishes that a specific aspect of caspofungin-induced Sko1 signaling is indeed to regulate the response to osmotic stress.

*YPD1* and *PTP2* inhibit HOG pathway signaling [43–45]. In *S. cerevisiae*, Sko1 increases *PTP2* expression following osmotic stress in order to control Hog1 [46]. We previously demonstrated that Hog1 does not regulate Sko1 following caspofungin stress but instead, phosphorylates Sko1 in response to NaCl stress [6]. Our results show increased Sko1 upstream intergenic region enrichment of *YPD1* and *PTP2* following caspofungin treatment. This finding suggests that Sko1 may inhibit Hog1 activity following cell wall stress. Interestingly, our results also show that Sko1 occupies its own upstream intergenic region (Fig 4G). This observation demonstrates that Sko1 likely regulates its own transcription following cell wall stress. We propose that self-regulation and inhibition of HOG pathway signaling may allow Sko1 to regulate osmotic stress response genes independently following caspofungin-induced cell wall stress.

## Extension of Sko1 cell wall related functions

Our ChIP-seq findings revealed new direct Sko1 target genes with functions in cell wall integrity, including *PGA56*, *PGA59*, *PGA62*, *PHR2*, and the chaperone *SSA2* (S2 Table). In a prior microarray study, Sko1 was found to modulate the expression of 11 cell wall genes in response to caspofungin treatment including *CRH11*, *PGA13*, and *PHR1* [6]. However, only one of these genes, *PGA31*, was identified as a direct Sko1 target gene in the current study. Taken together, these findings indicate that most Sko1-dependent cell wall integrity genes are indirectly regulated. We identified four transcription factor-encoding genes as direct Sko1 targets: *NRG1*, *EFG1*, *AHR1*, and *TCC1*. It is possible that these regulators work in concert with Sko1 to modulate CWD signaling. Chaperone function is essential for the cell wall adaptive response; for example, Hsp90 mediates caspofungin tolerance by modulating the PKC-MAPK

pathway [47]. However, the role of Ssa2 in the CWD response is unknown. Collectively, our ChIP-seq results and previous microarray findings extend our understanding of Sko1 control of cell wall integrity genes.

Our results also implicate a new role for Sko1 in cellular aggregation. We identified the transcriptional regulators, *EFG1* and *NRG1*, and the adhesin, *ALS1*, as Sko1 direct target genes (Fig 4A and S2 Appendix). Cellular aggregation is significantly increased following caspofungin treatment and is dependent on Als1 and Efg1 [48, 49], although the physiological basis of this behavior is unclear. Nevertheless, an *efg1Δ/Δ* mutant strain is hypersensitive to caspofungin and fails to upregulate *ALS1* [49]. The PKA-MAPK pathway regulates Efg1 in morphogenesis and biofilm formation but does not seem to play a role in the response to caspofungin treatment [49]. While a *sko1Δ/Δ* mutant strain does not have a defect in cellular adhesion and aggregation [50], Efg1 regulates multiple diverse processes, so it is likely that other regulators may indirectly control Efg1 activity in CWD signaling. Our findings reveal that Sko1 may play a role in caspofungin-induced Efg1/Als1-mediated cellular aggregation.

## Contribution of glycerol to cell wall stress adaptation

Our growth assay results indicate that overproduction of glycerol can partially suppress caspofungin hypersensitivity in a *sko1Δ/Δ* mutant strain (Fig 6A). Glycerol functions as a molecular protectant during hyperosmotic stress by replacing water as an intracellular solute and by maintaining turgor [27]. Also, glycerol is required to generate turgor during biofilm development and regulates the expression of several cell wall adhesin proteins including Als1 [51]. This finding strongly suggests that caspofungin hypersensitivity in the *sko1Δ/Δ* mutant strain is caused, in part, by the inability to counteract osmotic stress.

On the other hand, glycerol overproduction did not suppress caspofungin hypersensitivity in the *rlm1Δ/Δ* mutant strain (Fig 6B). This observation was surprising since *SKO1* overexpression partially suppressed this phenotype (Fig 5). Also, *rlm1Δ/Δ* mutant strains survive when grown on caspofungin medium containing 1.0 M sorbitol [12]. Therefore, other factors must contribute to caspofungin hypersensitivity in the *rlm1Δ/Δ* mutant strain. Our ChIP-seq results identified *PGA59*, *PGA62*, and *PHR2* as Rlm1 and Sko1 direct target genes, although we note that *PGA59* is bound by Rlm1 in both the presence and absence of caspofungin. We propose that *SKO1* overexpression suppresses caspofungin hypersensitivity in the *rlm1Δ/Δ* mutant strain by upregulating cell wall integrity genes.

## Conclusions

This study addressed four mechanistic aspects underlying the Sko1 and Rlm1 transcriptional response to caspofungin-induced cell wall signaling. (i) Sko1 and Rlm1 work cooperatively to regulate distinct gene classes whose functions include osmoregulation, adhesion, secretion, and glucose sensing. (ii) Sko1 specifically mediates the response to caspofungin-induced osmotic stress independently of HOG pathway signaling, Rlm1, and Cas5. (iii) *SKO1* caspofungin-induced transcription occurs via self-regulation, and (iv) glycerol accumulation can partially suppresses caspofungin hypersensitivity.

A key issue that remains unresolved is the mechanism through which Rlm1 activates Sko1 signaling. Our findings clearly demonstrate that this action occurs at the post-translational level. One hypothesis is that CWD-induced carbohydrate flux promotes increased nuclear accumulation of Sko1. In this context, we propose that *SHA3* and *HGT19* may promote Sko1 signaling. In *S. cerevisiae*, the Sha3 ortholog, Sks1, phosphorylates the Hgt19 ortholog, Itr1, to mediate glucose metabolism [39]. However, it is unknown if this interaction influences Sko1

activity. Exploring Sko1 function and localization in *sha3Δ/Δ* and *hgt19Δ/Δ* mutant strains would shed light on this idea.

## Methods

### Yeast strains and media

All *C. albicans* strains used in this study were derived from the wild-type reference strain BWP17 (*ura3Δ::λimm434/ura3Δ::λimm434, arg4::hisG/arg4::hisG, his1::hisG/his1::hisG*) or its derivative strains DAY185 and DAY286 [52, 53]. All strains used in experimental assays were isogenic, and genotypes are listed in S1 Table (S1 Table). Cultures were prepared in YPD$^{uri}$ (1% yeast extract, 2% peptone, 2% dextrose, 80 mg/L uridine) media at 30°C with shaking at 225 rpm unless otherwise noted.

### Construction of yeast strains

Overexpressing and epitope-tagged strains were constructed by PCR-mediated homologous recombination [52]. Oligonucleotide sequences are listed in S1 Table. Yeast transformations followed the lithium acetate protocol and all constructed strains were verified by colony PCR [54]. We created plasmid pCJN613 as a template to produce an amplicon containing V5-specific sequences. This plasmid contains the V5 epitope tag sequence from the pYES2.1/V5-His-TOPO vector (Invitrogen) and the nourseothricin resistance gene (*NAT*) sequence from plasmid pCJN542 [55]. The *RLM1-V5* amplicon was transformed into *C. albicans* strain DAY185 and integrated directly following the final *RLM1* amino acid-encoding codon to produce strain JMR228.

To construct the $P_{TDH3}SKO1$ and $P_{TDH3}SKO1-rlm1Δ/Δ$ overexpressing strains JMR171 and JMR213, we used pCJN542 as a template to produce an amplicon containing the glyceraldehyde-3-phosphate dehydrogenase *(TDH3)* promoter sequences, NAT resistance gene, and *SKO1*-specific sequences. The amplicon was transformed into the *C. albicans* wild-type strain (DAY185) or *rlm1Δ/Δ* mutant strain (VIC1090) and integrated directly upstream of the *SKO1* start codon.

To construct the $P_{TDH3}RHR2-sko1Δ/Δ$ and $P_{TDH3}RHR2-rlm1Δ/Δ$ overexpressing strains, JMR155 and SF001, we amplified the *TDH3* promoter and *NAT* resistance gene as described above. The amplicon was transformed into the *sko1Δ/Δ* mutant strain (JMR104) or *rlm1Δ/Δ* mutant strain (VIC1090) and integrated directly upstream of the *RHR2* start codon.

### Real-time-qPCR (RT-qPCR) assays

RT-qPCR reactions were performed as previously described [16]. Briefly, RNA was purified from log phase wild-type and *rlm1Δ/Δ* mutant cultures treated with or without 125 ng/mL caspofungin for 30 minutes using the RiboPure yeast kit (Ambion). cDNA was synthesized using the AffinityScript Multiple Temperature cDNA Synthesis Kit (Agilent Technologies) following the manufacturer's instructions. RT-qPCR reactions were prepared in triplicate and carried out at least twice using independently prepared yeast cultures. Gene expression changes were calculated using the $ΔΔC_T$ method [56]. Target gene fold changes for caspofungin-treated or untreated control groups were determined by comparison to the wild-type strain, and significant differences between groups were determined using unpaired t-tests.

### Genome-wide chromatin immunoprecipitation followed by sequencing (ChIP-seq)

We used the *SKO1-V5*, *RLM1-V5*, and isogenic wild-type reference strains for chromatin immunoprecipitation. Three independent samples were grown, processed, and sequenced for the tagged (experimental) and untagged (control) strains in the presence or absence of

caspofungin. Samples for ChIP-seq were harvested and processed as follows. Overnight cultures were diluted to an $OD_{600nm}$ of 0.2 in 200 mL of fresh $YPD^{uri}$ and grown to an $OD_{600nm}$ between 0.8 to 1.0. Next, cultures were split into two, and one set of cultures was exposed to 125 ng/mL of caspofungin for 30 minutes at 30˚C with shaking at 225 rpm. After 30 minutes, cells were harvested and processed for immunoprecipitation as described previously [57]. Briefly, crosslinking was performed for 15 min with formaldehyde. The crosslinks were quenched with glycine, and cells were lysed with glass beads using a Ribolyser sample homogenizer (Thermo Fisher). Chromatin was sheared into ~300 bp fragments by sonication using a Bioruptor (Diagenode). For each ChIP sample, immunoprecipitation was performed with 2 μg anti-V5 antibodies (Invitrogen) in a final volume of 500 μL with Protein A-Sepharose 4 Fast Flow beads (GE Healthcare). Following protease treatment and crosslinking reversal, samples were purified with phenol: chloroform and ethanol, air-dried and stored at -20˚C.

## Library preparation and sequencing

Library construction from reverse-crosslinked chromatin immunoprecipitated samples and DNA sequencing were performed by the DNA Technologies and Expression Analysis genome sequencing laboratory at the University of California, Davis. The Hyper DNA Library Prep Kit (KAPA) was used for library construction, and after size selection, samples were pooled and sequenced (40 bp, paired-end reads) using an Illumina NextSeq high-output 75 cycles (Specifications: 330–400 million CPF).

## ChIP-seq analysis

Illumina paired-end reads were trimmed using Trimmomatic (version 0.38) [58] to remove adapter sequences using the parameters ILLUMINACLIP:2:30:10 and leading or trailing low quality bases in reads with quality scores below 3. This was done while scanning the reads with a 4-base sliding window; reads were trimmed when the average quality dropped below 15. Additionally, read lengths below 36 bps were discarded. Trimmed, paired end reads were then aligned to the reference assembly (version 21) of the *C. albicans* genome (strain SC5314) obtained from the Candida Genome Database (CGD) [19]. Reference-based assembly was performed using Bowtie (version 1.2.2) [59], and reads that mapped at most to 3 genomic locations were randomly assigned to one of these loci. Maximum insert size for the paired-end reads was set to 1000. The peaks were visualized, and binding regions identified using MochiView [60] after converting bam files obtained using Bowtie to bigWig and wig formats using deepTools (version 3.3.0) [61].

## Consensus motif identification:

Peaks corresponding to Sko1 and Rlm1 binding events were identified and 500 bp regions extracted and centered around the identified peaks. The top 25% highly enriched peaks relative to the untagged control were used to identify the Sko1 consensus motif. MEME (version 5.1.1) [62] was used to identify motifs with the maximum width set to 40 bp (the reverse complement strand was also included for the motif search). The motifs obtained from MEME were corroborated with the motifs obtained using the motif finder function in MochiView [60]. Since fewer binding events were identified for Rlm1, the top 50% of highly enriched peaks relative to the untagged control were used to identify the Rlm1 consensus motif.

## Functional enrichment of target genes:

Cluster of Orthologous Groups (COG) [63] functional categories were obtained for all target genes of Rlm1 and Sko1 using eggNOG-mapper [64, 65]. The percentage of target genes

belonging to each functional category was compared to the percentage of all genes in the genome belonging to the same category. Gene Ontology (GO) enrichment [66] was also performed to identify enriched cellular components, biological processes and molecular functions in the target genes. GO terms were annotated through eggNOG-mapper and GO enrichment was performed using the topGO package [67]. To identify enriched GO terms, we used Fisher's exact test with a *P*-value cutoff of 0.01. Enriched KEGG pathways were identified using Pathview [68, 69].

## Growth assays

Growth assays on solid and liquid nutrient growth medium were performed as previously described [70]. Briefly, *C. albicans* overnight cultures were diluted to a starting $OD_{600nm}$ of 3.0. Cells were serially diluted and spotted onto $YPD^{uri}$ nutrient plates supplemented with 50 ng/mL or 125 ng/mL caspofungin (or without caspofungin as a control) to monitor growth under cell wall stress. Plates were incubated at 30˚C and photographed after 24 or 48 hours of growth. To examine growth kinetics under cell wall stress, cells were cultured overnight, standardized to an $OD_{600nm}$ of 0.2 in 200 μL $YPD^{uri}$ in a 96 well plate, and incubated with $YPD^{uri}$ supplemented with 50 ng/mL or 125 ng/mL caspofungin (or water as a control). Samples were grown for 18–24 hours at 30˚C with shaking, and $OD_{600nm}$ readings were acquired every 30 minutes using a Synergy Mx plate reader (Biotek).

## Supporting information

**S1 Table. Oligonucleotides and genotypes of strains used in this study.**
(XLSX)

**S2 Table. All Rlm1 and Sko1 direct target genes identified via ChIP-seq experiments, as well as GO Enrichment, COG, and *in silico* motif analyses for Rlm1 and Sko1 target gene upstream intergenic regions.** Also included are Sko1 target genes previously identified in caspofungin and osmotic stress microarray experiments.
(XLSX)

**S1 Fig. COG analysis of Sko1 and Rlm1 direct target genes.** COG functional categories were determined using eggNOG-mapper for A) Sko1 target genes and B) Rlm1 target genes.
(TIF)

**S2 Fig. Verification of the *SKO1* overexpressing strain.** Expression of Sko1 and Rlm1 CWD response genes, *PGA13*, *PGA31*, and *CRH11* was monitored via RT-qPCR analysis in the wild-type strain, *rlm1Δ/Δ* mutant strain and $P_{TDH3}SKO1$-*rlm1Δ/Δ* overexpressing strain. Gene expression changes were normalized to the caspofungin-treated wild-type strain adjusted to the value of 1.0. Data presented represents the mean of three biological replicates. The single asterisk indicates a *P*-value of ≤ 0.05 between the wild-type and *rlm1Δ/Δ* mutant strains. The double asterisk indicates a *P*-value of ≤ 0.05 between the wild-type and $P_{TDH3}SKO1$-*rlm1Δ/Δ* overexpressing strains.
(TIF)

**S3 Fig. KEGG analysis of Sko1 direct target genes.** Sko1 direct target genes are highlighted in red rectangular boxes.
(TIF)

**S1 Appendix. ChIP-seq MochiView plots of all Rlm1 direct target genes.** Genes are listed in descending order for Rlm1 upstream intergenic region enrichment.
(PDF)

**S2 Appendix. ChIP-seq MochiView plots of all Sko1 direct target genes.** Genes are listed in descending order for Sko1 upstream intergenic region enrichment.
(PDF)

## Acknowledgments

We thank Aaron Mitchell for the *rlm1Δ/Δ* mutant and *rlm1Δ/Δ/RLM1* complemented strains. We thank all members of the Rauceo and Nobile labs for insightful discussions on the manuscript. We also thank Peter Lipke for comments on the manuscript.

## Author Contributions

**Conceptualization:** Clarissa J. Nobile, Jason M. Rauceo.

**Data curation:** Marienela Y. Heredia, Mélanie A. C. Ikeh, Deepika Gunasekaran, Karen A. Conrad, Sviatlana Filimonava, Dawn H. Marotta, Jason M. Rauceo.

**Formal analysis:** Marienela Y. Heredia, Mélanie A. C. Ikeh, Deepika Gunasekaran, Karen A. Conrad, Sviatlana Filimonava, Dawn H. Marotta, Clarissa J. Nobile, Jason M. Rauceo.

**Funding acquisition:** Clarissa J. Nobile, Jason M. Rauceo.

**Investigation:** Marienela Y. Heredia, Mélanie A. C. Ikeh, Deepika Gunasekaran, Karen A. Conrad, Sviatlana Filimonava, Dawn H. Marotta, Jason M. Rauceo.

**Methodology:** Marienela Y. Heredia, Mélanie A. C. Ikeh, Deepika Gunasekaran, Karen A. Conrad, Sviatlana Filimonava, Dawn H. Marotta, Jason M. Rauceo.

**Project administration:** Clarissa J. Nobile, Jason M. Rauceo.

**Resources:** Clarissa J. Nobile, Jason M. Rauceo.

**Supervision:** Clarissa J. Nobile, Jason M. Rauceo.

**Validation:** Marienela Y. Heredia, Mélanie A. C. Ikeh, Deepika Gunasekaran, Karen A. Conrad, Sviatlana Filimonava, Dawn H. Marotta, Clarissa J. Nobile, Jason M. Rauceo.

**Visualization:** Marienela Y. Heredia, Deepika Gunasekaran, Karen A. Conrad, Sviatlana Filimonava, Dawn H. Marotta, Jason M. Rauceo.

**Writing – original draft:** Jason M. Rauceo.

**Writing – review & editing:** Marienela Y. Heredia, Mélanie A. C. Ikeh, Deepika Gunasekaran, Karen A. Conrad, Clarissa J. Nobile, Jason M. Rauceo.

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
