## [Decision Letter · Decision Letter 0]

13 May 2020

Dear Dr Rauceo,

Thank you very much for submitting your Research Article entitled 'An expanded cell wall damage signaling network is comprised of the transcription factors Rlm1 and Sko1 in Candida albicans' to PLOS Genetics. Your manuscript was fully evaluated at the editorial level and by independent peer reviewers. The reviewers were very positive, but they have suggested some minor corrections/clarifications that should be made before we can make a final decision on acceptance. In particular, please re-evaluate the data for 2 genes in Fig. 3 (see comments from Reviewer 2).

We therefore ask you to modify the manuscript according to the review recommendations before we can consider your manuscript for acceptance. Your revisions should address the specific points made by each reviewer.

[LINK]

Yours sincerely,

Geraldine Butler

Associate Editor

PLOS Genetics

Gregory P. Copenhaver

Editor-in-Chief

PLOS Genetics

Reviewer's Responses to Questions

**Comments to the Authors:**

Reviewer #1: The manuscript "An expanded cell wall damage signaling network is comprised of the transcription factors Rlm1 and Sko1 in Candida albicans" by Heredia et al., describes a link between cell wall damage pathways as well as novel biological roles. The research presented here is significant to the field as it enhances our understanding of the fungal response to antifungal agents, in this case, echinocandins. Additionally, a better understanding of these responses will likely lead to the identification of new drug targets. I am highly enthusiastic about the work presented and do not have significant comments.

Personally, I am interested in the cellular morphology and cell wall composition of the strains used in these studies and think some imaging would be beneficial to the reader and would aid in the understanding of the effect of these pathways on the overall cell structure and response. Additionally, it would be of interest to investigate the role of these pathways in the biofilm lifestyle of C. albicans and in non-albicans species. I also understand that due to the current situation, it is not plausible to acquire new data. This is just a suggestion and does not need to be incorporated unless the authors already have the data.

Reviewer #2: PGENETICS-D-20-00595; Heredia et al., ‘An expanded cell wall damage signaling network is comprised of the transcription factors Rlm1 and Sko1 in Candida albicans’

The authors present an interesting paper on signalling in response to cell wall stress in the human fungal pathogen Candida albicans, focusing on the transcription factors Rlm1 and Sko1. This work first demonstrates that Sko1 appears to be regulated by Rlm1 and contributes to its activity following cell wall damage. The authors then use ChIP-seq experiments in order to identify genes directly controlled by these transcription factors. The results from this work were generally more convincing for Sko1, as Rlm1 binding was only detected for a small number of genes and the peaks identified were generally more subtle. Potential consensus motifs for Rlm1 and Sko1 binding, based on the results from ChIP-seq, were then presented, before finally utilizing suppression analysis to provide further evidence of the link between these two transcription factors.

Key points

Fig 2, L247: The authors demonstrate a fairly constant (~40%) reduction in expression of Sko1 target genes in the rlm1 mutant. It would have been very useful in the figure to present the expression level of these genes in a sko1 mutant. This would have demonstrated the relative importance of Rlm1 and Sko1 in their regulation, and the interplay of these regulators in their control.

The results of ChIP-seq for Rlm1 only identified a very small set of potential target genes, as such the use of GO enrichment approaches is somewhat questionable. However, this analysis did potentially link Rlm1 function with the spindle pole and trafficking. This result could be discussed in relation to the known role of Slt2 in regulating polarised growth in budding yeast.

In Fig 3 the results presented for Pga59 and Sec10 are unconvincing, with no clear peak. In the discussion of the results it is also important to differentiate between genes that are potentially regulated in response to caspofungin or those that are bound irrespective of its addition. For example, the authors discuss Pga56 alongside Pga59 & Pga62, however, in the case of Pga56 its promoter appears to be bound equally well in the presence or absence of the drug. Finally, the authors identify Pga59 & Pga62 as potential targets, however, the binding site identified is identical in both cases plus appears to be more directly associated with orf19.2766.

The authors state they screened for the presence of the baker’s yeast Rlm1 binding motif in the promoter of C. albicans target genes (L257), however, no result of this is presented. Following on from this a highly diverged motif for Rlm1 binding in these two organisms is proposed, the authors could comment on how this fits with the conserved function of Rlm1 between these organisms and the likelihood of conserved targets.

Minor points

Please provide a reference or further information on the pHAS01 vector.

L125: should be ‘Sko1-specific sequences’

L:132 should be ‘upstream of the RHR2 start codon’.

Fig 1A, L229: Is the increase of Sko1 expression in the cas5 mutant significant?

Fig 1B, L238: Why was elevated expression of Sko1 seen in the complemented rlm1 mutant?

L303: ‘Highlighted a new biological role’ is an overstatement, without further evidence please refer to as ‘potential role’.

L307-325: What is the overlap between the 85 genes bound by Sko1 and the 26 originally described as caspofungin-inducible?

L318: Sko1 binds upstream of its own ORF in response to caspofungin. In addition, it also appears to bind across the ORF irrespective of the presence of caspofungin, the authors could comment on this.

L377 Does the motif for Sko1 identified relate to that previously reported for genes activated or repressed by Sko1?

Fis S2: Please provide stats analysis for Pga13 & Pga31, the same as that presented for Crh11.

Suppl. Table 2, Wksheet 2: For Rlm1 the top 50%, not 25%, enriched proteins were analysed according to the methods section.

Fig S4 L357: PGA13 is only one of multiple genes that could play a role, as such looking at only one potential gene did not really add to the manuscript.

**Have all data underlying the figures and results presented in the manuscript been provided?**

Reviewer #1: Yes

Reviewer #2: Yes

PLOS authors have the option to publish the peer review history of their article (what does this mean?). If published, this will include your full peer review and any attached files.

Reviewer #1: No

Reviewer #2: No

---

## [Editor Report · Decision Letter 1]

3 Jun 2020

Dear Dr Rauceo,

We are pleased to inform you that your manuscript entitled "An expanded cell wall damage signaling network is comprised of the transcription factors Rlm1 and Sko1 in Candida albicans" has been editorially accepted for publication in PLOS Genetics. Congratulations!

Yours sincerely,

Geraldine Butler

Associate Editor

PLOS Genetics

Gregory P. Copenhaver

Editor-in-Chief

PLOS Genetics

Comments from the reviewers (if applicable):

**Data Deposition**

http://datadryad.org/submit?journalID=pgenetics&manu=PGENETICS-D-20-00595R1

**Press Queries**

---

## [Editor Report · Acceptance letter]

1 Jul 2020

PGENETICS-D-20-00595R1 

An expanded cell wall damage signaling network is comprised of the transcription factors Rlm1 and Sko1 in <I>Candida albicans</I> 

Dear Dr Rauceo, 

We are pleased to inform you that your manuscript entitled "An expanded cell wall damage signaling network is comprised of the transcription factors Rlm1 and Sko1 in <I>Candida albicans</I>" has been formally accepted for publication in PLOS Genetics! Your manuscript is now with our production department and you will be notified of the publication date in due course.

With kind regards,

Kaitlin Butler

PLOS Genetics

On behalf of:
